# IS DEEP REINFORCEMENT LEARNING REALLY SUPER-HUMAN ON ATARI? LEVELING THE PLAYING FIELD

## ABSTRACT

Consistent and reproducible evaluation of Deep Reinforcement Learning (DRL) is not straightforward. In the Arcade Learning Environment (ALE), small changes in environment parameters such as stochasticity or the maximum allowed play time can lead to very different performance. In this work, we discuss the difficulties of comparing different agents trained on ALE. In order to take a step further towards reproducible and comparable DRL, we introduce SABER, a **S**tandardized **A**tari **BE**nchmark for general **R**einforcement learning algorithms. Our methodology extends previous recommendations and contains a complete set of environment parameters as well as train and test procedures. We then use SABER to evaluate the current state of the art, Rainbow. Furthermore, we introduce a *human world records baseline*, and argue that previous claims of *expert or superhuman* performance of DRL might not be accurate. Finally, we propose *Rainbow-IQN* by extending Rainbow with Implicit Quantile Networks (IQN) leading to new state-of-the-art performance. Source code is available for reproducibility.

## 1 INTRODUCTION

Human intelligence is able to solve many tasks of different natures. In pursuit of generality in artificial intelligence, video games have become an important testing ground: they require a wide set of skills such as perception, exploration and control. Reinforcement Learning (RL) is at the forefront of this development, especially when combined with deep neural networks in DRL. One of the first general approaches reaching reasonable performance on many Atari games while using the exact same hyper-parameters and neural network architecture was Deep Q-Network (DQN) (Mnih et al., 2015), a value based DRL algorithm which directly takes the raw image as input. This success sparked a lot of research aiming to create better, faster and more stable general algorithms. The ALE (Bellemare et al., 2013), featuring more than 60 Atari games (see Figure 1), is heavily used in this context. It provides many different tasks ranging from simple paddle control in the ball game Pong to complex labyrinth exploration in Montezuma's Revenge which remains unsolved by general algorithms up to today.

As the number of contributions is growing fast, it becomes harder and harder to make a proper comparison between different algorithms. In particular, a relevant difference in the training and evaluation procedures exists between available publications. Those issues are exacerbated by the fact that training DRL agents is very time consuming, resulting in a high barrier for reevaluation of previous work. Specifically, even though ALE is fast at runtime, training an agent on one game takes approximately one week on one GPU and thus the equivalent of more than one year to train on all 61 Atari games. A standardization of the evaluation procedure is needed to make DRL *that matters* as pointed out by Henderson et al. (2018) for the Mujoco benchmark (Todorov et al., 2012): the authors criticize the lack of reproducibility and discuss how to allow for a fair comparison in DRL that is consistent between articles.

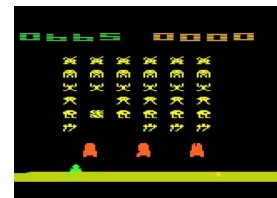

Figure 1: ALE Space Invaders

In this work, we first discuss current issues in the evaluation procedure of different DRL algorithms on ALE and their impact. We then propose an improved evaluation procedure, extending the recommendations of Machado et al. (2018), named SABER : a Standardized Atari BEnchmark for

Reinforcement learning. We suggest benchmarking on the *world records human baseline* and show that RL algorithms are in fact far from solving most of the Atari games. As an illustration of SABER, current state-of-the-art DRL algorithm Rainbow (Hessel et al., 2018) is benchmarked. Finally, we introduce and benchmark on SABER a new state-of-the-art agent: a distributable combination of Rainbow and Implicit Quantiles Network (IQN) (Dabney et al., 2018).

The main contributions of this work are :

- The proposal, description and justification of the SABER benchmark.

- Introduction of a *world records human baseline*. We argue it is more representative of the human level than the one used in most of previous works. With this metric, we show that the Atari benchmark is in fact a hard task for current general algorithm.

- A SABER compliant evaluation of current state-of-the art agent Rainbow.

- A new state-of-the-art agent on Atari, Rainbow-IQN, with a comparison on SABER to Rainbow, to give an improvement range for future comparisons.

- For reproducibility sake, an open-source implementation of Rainbow, Rainbow-IQN, distributed following the idea from Horgan et al. (2018).

## 1.1 RELATED WORK

**Reproducibility and comparison in DRL**   *Deep Reinforcement Learning that matters* (Henderson et al., 2018) is one of the first works to warn about a reproducibility crisis in the field of DRL. This article relies on the MuJoCo (Todorov et al., 2012) benchmark to illustrate how some common practices can bias reported results. As a continuation to the work of Henderson et al. (2018), J. Pineau introduced a Machine Learning reproducibility checklist (Pineau, 2019) to allow for reproducibility and fair comparison. Machado et al. (2018) deal with the Atari benchmark. They describe the divergence in training and evaluation procedures and how this could lead to difficulties to compare different algorithms. A first set of recommendations to standardize them is introduced, constituting the basis of this work and will be summarized in the next section. Finally, the Github Dopamine (Castro et al., 2018) provides an open-source implementation of some of the current state-of-the-art algorithms on Atari benchmark, including Rainbow and IQN. An evaluation following almost all guidelines from Machado et al. (2018) are provided in Castro et al. (2018). However the implementation of Rainbow is partial, and the recommendation of using the full action set is not applied. This is why our work contains a new evaluation of Rainbow.

**Value based RL**   DQN (Mnih et al., 2015) is the first value based DRL algorithm benchmarked on all Atari games with the exact same set of hyperparameters (although previous work by Hausknecht et al. (2014) already performed such a benchmark with neural networks). This algorithm relies on the well known Q-Learning algorithm (Watkins & Dayan, 1992) and incorporates a neural network. Deep Q-learning is quite unstable and the main success of this work is to introduce practical tricks to make it converge. Mainly, transitions are stored in a *replay memory* and sampled to avoid correlation in training batch, and a separate *target network* is used to avoid oscillations. Since then, DQN has been improved and extended to make it more robust, faster and better. Rainbow (Hessel et al., 2018) is the combination of 6 of these improvements (Van Hasselt et al., 2016; Schaul et al., 2015; Bellemare et al., 2017; Wang et al., 2015; Fortunato et al., 2017; Mnih et al., 2016) implemented in a single algorithm. Some ablations studies showed that the most important components were Prioritized Experience Replay (PER) (Schaul et al., 2015) and C51 (Bellemare et al., 2017). The idea behind PER is to sample transitions according to their *surprise*, i.e. the worse the network is at predicting the Q-value of a specific transition, the more we sample it. C51 is the first algorithm in *Distributional RL* which predicts the full distribution of the Q-function instead of predicting only the mean of it. Finally, IQN (Dabney et al., 2018) is an improvement over C51. It almost reaches on its own the performance of the full Rainbow with all 6 components. In C51 the distribution of the Q-function is represented as a categorical distribution while in IQN, it is represented by implicit quantiles.

## 2 CHALLENGES WHEN COMPARING PERFORMANCE ON THE ATARI BENCHMARK

In this section we discuss several challenges to make a proper comparison between different algorithms trained on the Atari benchmark. First, we briefly summarize the initial problems and their solution as proposed by Machado et al. (2018). Then we detail a remaining issue not handled by those initial standards, the maximum length time allowed for an episode. Finally, we introduce a readable metric, representative of actual human level and allowing meaningful comparison.

### 2.1 REVISITING ALE: AN INITIAL STEP TOWARDS STANDARDIZATION

Machado et al. (2018) discuss about divergence of training and evaluation procedures on Atari. They show how those divergences are making comparison extremely difficult. They establish recommendations that should be used in order to standardize the evaluation process.

**Stochasticity** The ALE environment is fully deterministic, i.e. leading to the exact same state if the exact same actions are taken at each state. This is actually an issue for general algorithm evaluation. For example, an algorithm learning *by heart* good trajectories can actually reach a high score with an open-loop behaviour. To handle this issue, Machado et al. (2018) introduce *sticky actions*: actions coming from the agent are repeated with a given probability $\xi$, leading to a non deterministic behavior. They show that sticky actions are drastically affecting performance of an algorithm exploiting the environment determinism without hurting algorithms learning more robust policies like DQN (Mnih et al., 2015). We use sticky actions with probability $\xi = 0.25$ (Machado et al., 2018) in all our experiments.

**End of an episode: Use actual game over** In most of the Atari games the player has multiple lives and the game is actually over when all lives are lost. But some articles, e.g. DQN, Rainbow, IQN, end a training episode after the loss of the first life but still use the standard game over signal while testing. This can in fact help the agent to learn how to avoid death and is an unfair comparison to agents which are not using this *game-specific* knowledge. Machado et al. (2018) recommend to use only the standard game over signal for all games while training.

**Action set** Following the recommendation of Machado et al. (2018) we do not use the *minimal useful action set* (the set of actions having an effective impact on the current game) as used by many previous works (Mnih et al., 2015; Hessel et al., 2018). Instead we always use all 18 possible actions on the Atari Console. This removes some specific domain knowledge and reduces the complexity of reproducibility. For some games, the minimal useful action set is different from one version to another of the standard Atari library: an issue to reproduce result on breakout was coming from this (Graetz, 2018).

**Reporting performance** As proposed by Machado et al. (2018), we report our score while training by averaging $k$ consecutive episodes (we have set $k = 100$). This gives information about the stability of the training and removes the statistical bias induced when reporting score of the best policy which is today a common practice (Mnih et al., 2015; Hessel et al., 2018).

### 2.2 MAXIMUM EPISODE LENGTH

A major parameter is left out of the work of Machado et al. (2018): the maximum number of frames allowed per episode. This parameter ends the episode after a fixed number of time steps even if the game is not over. In most of recent works (Hessel et al., 2018; Dabney et al., 2018), this is set to 30 min of game play and only to 5 min in Revisiting ALE (Machado et al., 2018). This means that the reported scores can not be compared fairly. For example, in easy games (e.g. Atlantis, Enduro), the agent never dies and the score is more or less linear to the allowed time: the reported score will be 6 times higher if capped at 30 minutes instead of 5 minutes.

We argue that the time cap can make the performance comparison non significant. On many games (e.g. Atlantis, Video Pinball) the scores reported of Ape-X (Horgan et al., 2018), Rainbow (Hessel et al., 2018) and IQN (Dabney et al., 2018) are almost exactly the same. This is because all agents

reach the time limit and get the highest possible score in 30 minutes: the difference in scores is due to minor variations, not algorithmic difference. As a consequence, the more successful agents are, the more games are incomparable because they reach the maximum possible score in the time cap.

This parameter can also be a source of ambiguity and error. The best score on Atlantis (2,311,815) is reported by *Proximal Policy Optimization* by Schulman et al. (2017) but this score is almost certainly wrong: it seems impossible to reach it in only 30 minutes! The first distributional paper, C51 (Bellemare et al., 2017), also did this mistake and reported wrong results before adding an erratum in a later version on ArXiv.

We argue that episodes should not be capped at all. The original ALE article (Bellemare et al., 2013, pg.3) states that *This functionality is needed for a small number of games to ensure that they always terminate*. On some famously hard games like *Pitfall* and *Tennis*, random exploration leads to much more negative reward than positive and thus the agent effectively learns to do nothing, e.g. not serving in Tennis. We claim that, even with this constraint, agents still end up learning to do nothing, and the drawback of the cap harms the evaluation of all other games. Moreover, the human high scores for Atari games have been achieved in several hours of play, and would have been unreachable if limited to 30 minutes.

To summarize, ideally one would not cap at all length of episode while training and testing. However this makes some limitations of the ALE environment appear, as described in the following paragraph.

**Glitch and bug in the ALE environment**

When setting the maximum length of an episode to infinite time, the agent gets stuck on some games, i.e. the episode never ends, because of a bug in the emulator. In this case, even doing random actions for more than 20 hours neither gives any reward nor end the game. This happens consistently on *BattleZone* and less frequently on *Yar's Revenge*. One unmanaged occurrence of this problem is enough to hamper the whole training of the agent. It is important to note that those bugs were discovered by chance and it is probable that this could happen on some other games.

We recommend to set the maximum episode length to infinite (in practice, a limit of 100 hours was used). Additionally we suggest a *maximum stuck time* of 5 minutes. Instead of limiting the allowed time for the agent, we limit the time without receiving any reward. This small trick handles all issues exposed above, and sets all reported scores on the same basis, making comparison to world records possible.

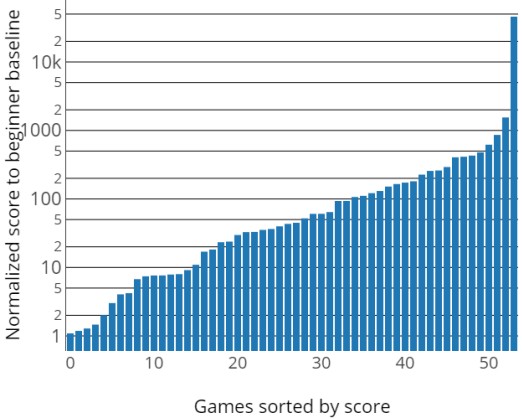

Figure 2: World records scores vs. the usual beginner human baseline (Mnih et al., 2015) (log scale).

Other bugs or particularities harming evaluation were encountered while training on the full Atari benchmark: buffer rollover with sudden negative score, influence of a start key for some games, etc. They are detailed and discussed in the supplementary material and we argue that they can have a drastic impact on performance and explain inconsistencies.

## 2.3 HUMAN WORLD RECORDS BASELINE

A common way to evaluate AI for games is to let agents compete against human world champions. Recent examples for DRL include the victory of AlphaGo versus Lee Sedol for Go (Silver et al., 2016), OpenAI Five on Dota 2 (OpenAI, 2018) or AlphaStar versus Mana for StarCraft 2 (Vinyals et al., 2019). In the same spirit, one of the most used metric for evaluating RL agents on Atari is to compare them to the human baseline introduced by Mnih et al. (2015). Previous works use the normalized human score, i.e. 0% is the score of a random player and 100% is the score of the human baseline, which allows to summarize the performance on the whole Atari set in one number, instead of individually comparing raw scores for each of the 61 games. However we argue

that this human baseline is far from being representative of the best human player, which means that using it to claim superhuman performance is misleading. The current world records are available online for 58 of the 61 evaluated Atari game [1]. Evaluating these world records scores using the usual human normalized score has a median of 4.4k% and a mean of 99.3k% (see Figure 2 for details), to be compared to 200% and 800% of original Rainbow (Hessel et al., 2018). As a consequence, we argue that using a normalized human score with the world records will give a much better indication of the performance of the agents and the margin of improvement. Note that 3 games of the ALE (double dunk, elevator action and tennis) do not have a registered world record, so all following experiments contain 58 games.

## 3 SABER : A STANDARDIZED ATARI BENCHMARK FOR REINFORCEMENT LEARNING

In this section we introduce SABER, a set of training and evaluation procedures on the Atari benchmark allowing for fair comparison and for reproducibility. Moreover, those procedures make it possible to compare with the human world records baseline introduced above and thus to obtain an accurate idea of the gap between general agents and best human players.

### 3.1 TRAINING AND EVALUATION PROCEDURES

All recommendations stated in the previous section are summarized in Table 1 to constitute the SABER benchmark. It is important to note that those procedures must be used at both training and test time. The recent work Go-Explore (Ecoffet et al., 2019) opened a debate on allowing or not stochasticity at training time. They report state-of-the-art performance on the famously hard game *Montezuma's Revenge* by removing stochasticity at training time. They conclude that we should have benchmarks with and without it (Ecoffet Adrien & Clune, 2018). We choose to use same conditions for training and testing general agents: this is more in line with realistic tasks.

### 3.2 REPORTING RESULTS

Table 1: Game parameters of SABER

| Parameter | Value |
|---|---|
| Sticky actions | $\xi = 0.25$ |
| Life information | Not allowed |
| Action set | 18 actions |
| Max stuck time | 5 min (18000 frames) |
| Max episode length | Infinite (100 hours) |
| Initial state and random seed | Same starting state and varying seed |

In accordance with previous guidelines, we advocate to report mean scores of 100 consecutive training episodes at specific time, here 10M, 50M, 100M and 200M frames. This removes the bias of reporting scores of the best agent encountered during training and makes it possible to compare at different data regimes. Due to the complexity of comparing 58 scores in a synthetic manner, we try to provide a single metric to make an effective comparison. Mean and median normalized scores to the records baseline are computed over all games. Note that the median is more relevant: the mean is highly impacted by outliers, in particular by games where the performance is superhuman. For the mean value, games with an infinite game time and score are artificially capped to 200% of the records baseline. We propose to add a histogram of the normalized score, to classify the games according to their performance. We define 5 classes: failing ($< 1\%$), poor ($< 10\%$), medium ($< 50\%$), fair ($< 100\%$) and superhuman ($> 100\%$). Medians, means and histograms can be found in Section 5, and the fully detailed scores are available in the supplementary materials.

## 4 RAINBOW-IQN

Two different approaches were combined to obtain an improvement over Rainbow (Hessel et al., 2018): Rainbow itself and IQN (Dabney et al., 2018) because of its excellent performance. Im-

---

[1]on the TwinGalaxies website https://www.twingalaxies.com/games.php?platformid=5

plementation details and hyper-parameters are described in the supplementary material. Both our implementations of Rainbow and Rainbow-IQN are distributed [2], following Ape-X (Horgan et al., 2018) and based on the implementation of (Castro et al., 2018).

IQN is an evolution of the C51 algorithm (Bellemare et al., 2017) which is one of the 6 components of the full Rainbow, so this is a natural upgrade. After the implementation, preliminary tests highlighted the impact of PER (Schaul et al., 2015): taking the initial hyper-parameters for PER from Rainbow resulted in poor performance. Transitions are sampled from the replay memory proportionally to the training loss to the power of priority exponent $\omega$. Reviewing the distribution of the loss shows that it is significantly more spread for Rainbow-IQN than for Rainbow, thus making the training unstable, because some transitions were over-sampled. To handle this issue, 4 values of $\omega$ were tested on 5 games: 0.1, 0.15, 0.2, 0.25 instead of 0.5 for original Rainbow, with 0.2 giving the best performance. The 5 games were Alien, Battle Zone, Chopper Command, Gopher and Space Invaders. All other parameters were left as is. Rainbow-IQN is evaluated on SABER and compared to Rainbow in the following section.

## 5 EXPERIMENTS

In this section, we describe the experiments performed on SABER. For all parameters not mentioned in SABER (e.g. the action repeat, the network architecture, the image preprocessing, etc) we carefully followed the parameters used in Rainbow (Hessel et al., 2018) and IQN (Dabney et al., 2018) papers. Those details and the scores for each agent and individual games can be found in the supplementary materials. Training one agent takes slightly less than a week, which makes a full benchmark use around 1 year-GPU. As a consequence, for each algorithm benchmark, trainings were run with only one seed for the full benchmark, and 5 seeds for 14 of the 61 games. Details on the choice of these games and the associated scores can be found in Section 5.3. The combined duration of all experiments conducted for this article is more than 4 years-GPU. Agents are trained using SABER guidelines on the 61 Atari games, and evaluated with the records baseline for 58 games. Scores at both 5 minutes and 30 minutes are kept while training to compare to previous works.

### 5.1 RAINBOW EVALUATION

| Algorithm | Original Rainbow (Hessel et al., 2018) | | | Following (Machado et al., 2018) | | |
|---|---|---|---|---|---|---|
| | Median | Mean | Superhuman | Median | Mean | Superhuman |
| Performance | 4.20% | 24.10% | 2 | 2.61% | 17.09% | 1 |

Table 2: Median and mean human-normalized performance and number of superhuman scores ($> 100\%$). Scores are coming from the original Rainbow and from our re-evaluation of Rainbow following recommendations of Machado et al. (30 minutes evaluation, at 200M training frames).

Benchmarking Rainbow makes it possible to measure the impact of the guidelines of Machado et al.: sticky actions, ignore life signal and full action set. Table 2 compares the originally reported performance of Rainbow (Hessel et al., 2018) to an evaluation following the recommendations of Machado et al. The performance is measured with the records baseline, for a 30 minutes evaluation at 200M training frames, to be as close as possible to the conditions of the original Rainbow. The impact of the standardized training procedure is major: as shown in the following paragraph, the difference in median (1.59%) is comparable to the difference between DQN and Rainbow (1.8%, see Figure 5) when both are trained on same training procedures. This demonstrates the importance of explicit and standardized training and evaluation procedures.

### 5.2 RAINBOW-IQN: EVALUATION AND COMPARISON

**Influence of maximum episode length** Table 3 studies the influence of the time limit for the evaluation, by reporting performance for Rainbow and Rainbow-IQN depending on the evaluation time. A significant difference can be seen between 5, 30 minutes and without limiting time of evaluation, which confirms the discussion of Section 2.2.

---

[2]See supplementary materials for details

| Time | 5 min | | | 30 min | | | No limit (SABER) | | |
|---|---|---|---|---|---|---|---|---|---|
| | Median | Mean | Super. | Median | Mean | Super. | Median | Mean | Super. |
| Rainbow | 2.35% | 14.86% | 0 | 2.61% | 17.09% | 1 | 2.83% | 24.54% | 3 |
| Rainbow-IQN | 2.61% | 17.62% | 0 | 2.81% | 20.18% | 1 | 3.13% | 30.89% | 4 |

Table 3: Evolution of performance with evaluation time (mean, median of normalized baseline and number of superhuman agents) for Rainbow and Rainbow-IQN.

### Median performance with regards to training frames

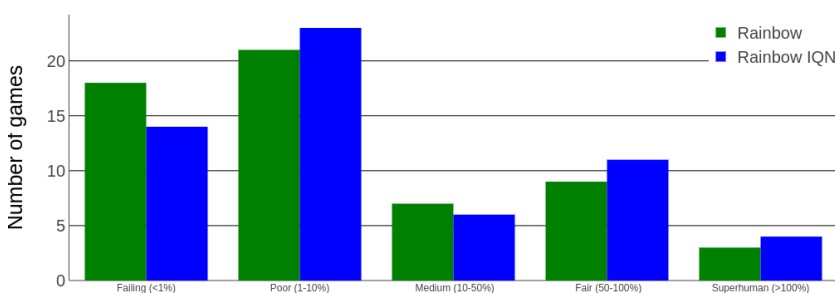

Figure 3: Comparison of Rainbow and Rainbow-IQN on SABER: Median normalized scores with regards to training steps.

**Comparison to Rainbow**    As introduced in Section 3.2, we compare Rainbow and Rainbow-IQN with median and mean metrics on SABER conditions, and with a classification of the performance of the agents in Figure 4. Figure 3 shows that Rainbow-IQN performance during training is consistently higher than Rainbow. One can notice on Figure 4 that the majority of agents are in the *poor* and *failing* categories, showing the gap that must be crossed to achieve superhuman performance on the ALE.

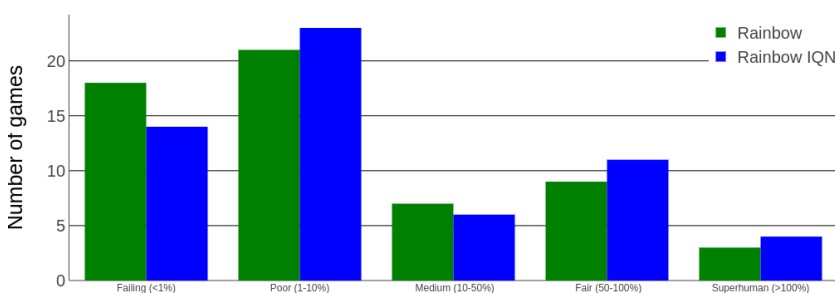

Figure 4: Comparison of Rainbow and Rainbow-IQN on SABER: classifying performance of agents relatively to the records baseline (at 200M training frames).

**Comparison to DQN**    Figure 5 provides a comparison between DQN, Rainbow and Rainbow-IQN. The evaluation time is set at 5 minutes to be consistent with the reported score of DQN by Machado et al. (2018). As expected, DQN is outperformed for all training steps. As aforementioned,

the difference between DQN and Rainbow is in the same range as the difference coming from divergent training procedures, showing again the necessity for standardization.

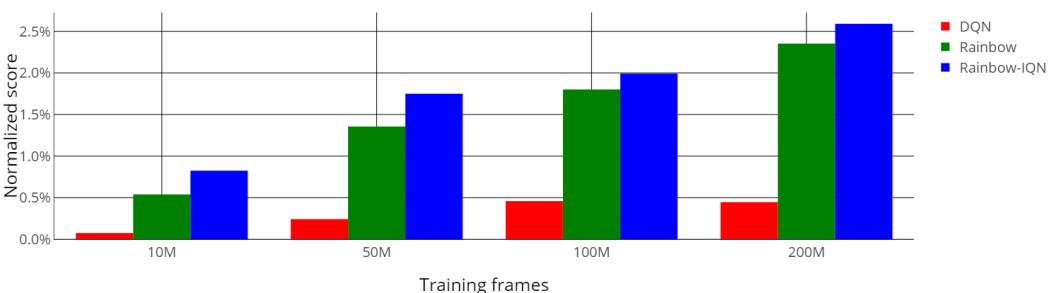

Figure 5: Median performance comparison for DQN, Rainbow and Rainbow-IQN with regards to training frames. Evaluation time is set at 5 minutes to allow a comparison to DQN.

### 5.3    STABILITY OF BOTH RAINBOW AND RAINBOW-IQN

Machado et al. (2018) use 5 different seeds for training to check that the results are reproducible and stable. For this article, these 5 runs are conducted on both Rainbow and Rainbow-IQN for 14 games (around 25% of the whole benchmark). It would be best to have the whole benchmark on 5 seeds but this was way above our computational resources. Still, these 14 games allow us to make a first step of stability studies. They are chosen according to the results of the first seed, with the idea of prioritizing games on which scores were most notably different between Rainbow and Rainbow-IQN. We also try to choose diverse games from different categories (from failing to superhuman) and we removed the 5 games used for the hyperparameter tuning. Games that were either too hard (such as *Montezuma's Revenge* or *Pitfall*) or too simple (such as *Pong* or *Atlantis*) are intentionally excluded to make the additional tests as significant as possible. For each game with 5 seeds conducted, we computed the median and mean human-normalized performance averaged over the 5 trials. This way, we can both have a reasonable estimation of the stability of the trainings, and a comparison as fair as possible between Rainbow and Rainbow-IQN.

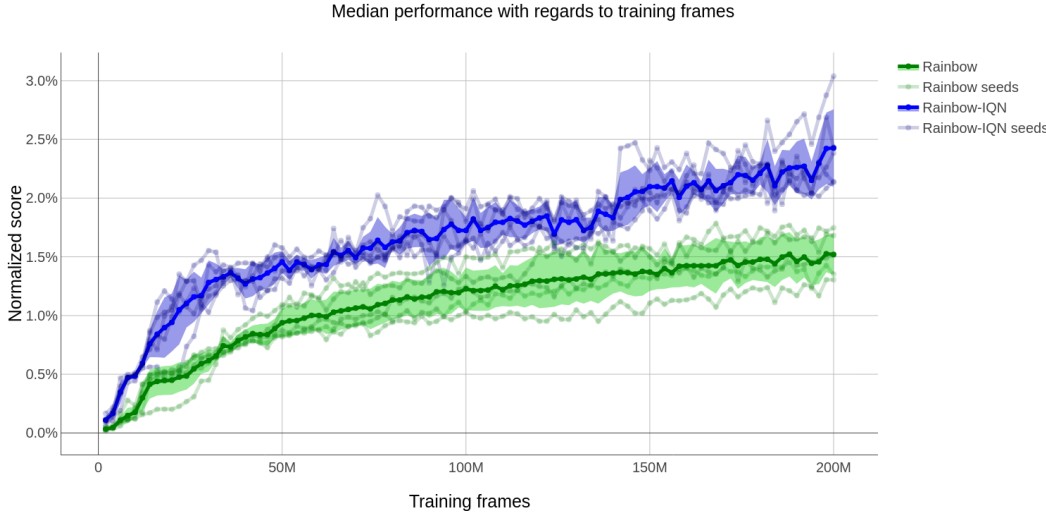

Figure 6: Median normalized scores with regards to training steps averaged over 5 seeds for both Rainbow and Rainbow-IQN. Only the 14 games on which 5 seeds have been conducted were used for this figure.

Figure 6 shows the median averaged over 5 trials for both Rainbow and Rainbow-IQN. We also plot each seed separately and the standard deviation over the 5 seeds. This has been computed only on

the 14 games on which we succeeded to conduct 5 runs. This shows that standard deviations are roughly similar for Rainbow and Rainbow-IQN, around 0.2 % on the world record baseline. As these standard deviations are rather small for 25% of the Atari games, we can assume they would be still small on the whole benchmark. We think that this reveals that both Rainbow and Rainbow-IQN are quite stable on Atari and strengthens our confidence on Rainbow-IQN being the new state-of-the-art on the Atari benchmark. In particular, Rainbow-IQN reaches *infinite game time* on *Asteroids* on all 5 trials whereas Rainbow fails for each seed.

# 6    Conclusion: why is RL that Bad at Atari Games?

In the current work, we confirm the impact of standardized guidelines for DRL evaluation, and build a consolidated benchmark, SABER. The importance of the play time is highlighted: agents should be trained and evaluated with no time limitation. To provide a more significant comparison, a new baseline is built, based on human world records. Following these recommendations, we show that the state-of-the-art Rainbow agent is in fact far from human world records performance. As a further illustration, we provide an improvement, Rainbow-IQN, and use it to measure the impact of the evaluation time over performance.

The striking information from these results is that general DRL algorithms are far from best human performance. The median of world records human normalized score for Rainbow-IQN is 3,1%, meaning that for half of the games, the agent is only 3% of the way from random play to the actual best human play. There are many possible reasons for this failure, which we will briefly discuss here to give an intuition of the current limitations of general DRL algorithm.

**Reward clipping**    In some games the optimal play for the RL algorithm is not the same as for the human player. Indeed, all rewards are clipped between -1 and 1 so RL agents will prefer to obtain many small rewards over a single large one. This problem is well represented in the game *Bowling*: the agent learns to avoid striking or sparing. Indeed the actual optimal play is to perform 10 strikes in a row leading to one big reward of 300 (clipped to 1 for the RL agent) but the optimal play for the RL agent is to knock off bowling pins one by one. This shows the need of a better way to handle reward of different magnitude, by using an invertible value function as suggested by Pohlen et al. (2018) or using Pop-Art normalization (van Hasselt et al., 2016).

**Exploration**    Another common reason for failure is a lack of exploration, resulting in the agent getting stuck in a local minimum. Random exploration or Noisy Networks (Fortunato et al., 2017) are far from being enough to solve most of Atari games. In *Kangaroo* for example, the agent learns to obtain rewards easily on the first level but never tries to go to the next level. This problem might be exacerbated by the reward clipping: changing level may yield a higher reward, but for the RL algorithm all rewards are the same. Exploration is one of the most studied field in Reinforcement Learning, so possible solutions could rely on curiosity (Pathak et al., 2017) or count-based exploration (Ostrovski et al., 2017).

**Human basic knowledge**    Atari games are designed for human players, so they rely on implicit prior knowledge. This will give a human player information on actions that are probably positive, but with no immediate score reward (climbing a ladder, avoiding a skull etc). The most representative example can be seen in *Riverraid*: shooting a fuel container gives an immediate score reward, but taking it makes it possible to play longer. Current general RL agents do not identify it as a potential bonus, and so die quickly. Even with smart exploration, this remains an open challenge for any general agent.

**Loop on a sub-optimal policy**    Finally, we discovered that on some games the agent finds quickly a loop continuously giving a small amount of reward and spends the whole training on this loop. In *Bank Heist* for example, the agent understood that bonus were respawning when changing level. Therefore the agent learned to just take over and over the same bonus until game timeout, failing to reach a good score. A very similar behaviour was discovered on *Elevator Action,Kangaroo, Krull* and *Tutankham*.

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

## A  SUPPLEMENTARY MATERIALS: IMPLEMENTATION DETAILS

### A.1  RAINBOW APE-X

Practically, we started with the PyTorch (Paszke et al., 2017) open source implementation of Rainbow coming from Kaikhin (Kaixhin, 2018). We tested this initial implementation on some games with the exact same training conditions as in the original Rainbow to ensure our results were consistent. After this sanity check, we implemented a distributed version of Rainbow following the paper Distributed Prioritized Experience Replay (Ape-X) (Horgan et al., 2018). Ape-X (Horgan et al., 2018) is a distributed version of Prioritized Experience Replay (PER) but which can be adapted on any value-based RL algorithm including PER, e.g. Rainbow. There is no study of this in the main article because we lacked time and computing resources to run experiments on whole Atari set with distributed actors. However, some experiments were conducted to ensure our distributed implementation was working as expected. These experiments are detailed in the next section. We claim that our Ape-X implementation is an important practical improvement compared to the single

agent implementation of both Dopamine (Castro et al., 2018) and Kaikhin (Kaixhin, 2018). It is important to note that all the experiments detailed in the main paper have been made with a single actor and thus do not really show the interest of distributed Rainbow Ape-X. A lock was added to synchronize all single-agent experiments to ensure that one step of learner is done every 4 steps of actor as in the original Rainbow (Hessel et al., 2018). All our hyperparameter values match closely those reported in Rainbow (Hessel et al., 2018). There is still one difference coming from our Ape-X implementation (even using a single actor). Indeed, we compute priorities before putting transitions in memory instead of putting new transitions with the maximum priorities seen as in the original Rainbow (Hessel et al., 2018). We argue that this should not have much impact on single-actor setting and that it is straightforward to implement for each algorithm using Prioritized Experience Replay (Schaul et al., 2015).

For the distributed memory implementation, we use a key-memory database with REDIS (Redis, 2019). The database is kept in RAM, which makes access faster and is possible for the ALE considering the size of the images and the replay memory size.

### A.2 Rainbow-IQN Ape-X

We combined our Rainbow Ape-X implementation with IQN (Dabney et al., 2018) coming from the TensorFlow (Abadi et al., 2016) open source implementation of Dopamine (Castro et al., 2018) to obtain a PyTorch (Paszke et al., 2017) implementation of *Rainbow-IQN Ape-X*. All our hyperparameter values match closely those reported in IQN. As indicated in the main paper, we had to tune the *priority exponent* coming from Prioritized Experience Replay (Schaul et al., 2015) in order to make the training stable. We tested both value of learning rate and epsilon of the adam optimizer from Rainbow and from IQN. A minor improvement in performance was found with the learning rate of IQN (Dabney et al., 2018) (tested only on 3 games for computational reasons), which was then used for all our experiments.

## B Experiments

### B.1 Image preprocessing and architecture

We used the same preprocessing procedure used in Rainbow and IQN, i.e an action repeat of 4, frames are converted to grayscale, resized to 84*84 with a bilinear interpolation [3] and max-pooled over 2 adjacent frames. The actual input to our network consists in 4 stacked frames.

Our architecture followed carefully the one from the original DQN for the main branch which was also used in Rainbow and IQN. The branch responsible of implicit quantiles is made exactly as the one from the original implementation section of IQN (Dabney et al., 2018, p.g. 5)

### B.2 Training infrastructure

The training of the agents was split over several computers and GPUs, containing in total:

- 3 Nvidia Titan X and 1 Nvidia Titan V (training computer)
- 1 Nvidia 1080 Ti (local workstation)
- 2 Nvidia 1080 (local workstations)
- 3 Nvidia 2080 (training computer)
- 4 Nvidia P100 (in a remote supercomputer)
- 2 Nvidia V100 (in a remote supercomputer)
- 4 Nvidia Tesla V-100 (DGX station)
- 4 Nvidia Quadro M2000 (local workstations)

---

[3]for some experiments we made this interpolation using the Python image library PIL instead of OpenCV because OpenCV was not available on the remote supercomputer. This was leading to small differences in the final resized image.

### B.3 RAINBOW-IQN APE-X

To ascertain our distributed implementation of Rainbow-IQN was functional, 3 experiments were conducted with multiple actors (10 actors instead of one). All locks and synchronization processes are removed to let actors fill the replay memory as fast as possible. The experiments are stopped when the learner reaches the same number of steps as in our single-agent experiments.

Table 4 reports the raw scores obtained by the agents on the selected games. Although the same number of batches is used in the training, there is a huge improvement in performances for the 3 games tested over the single agent version. This confirms the results coming from the Ape-X (Horgan et al., 2018) paper. Even at same learner step, the agent can benefit greatly from more experiences coming from multiple actors. Thanks to PER, the learner focuses on the most important transitions in the replay memory. Moreover this could avoid being stuck in a local minimum as assumed in Ape-X (Horgan et al., 2018). For the 3 experiments done, all actors together played around 6 times more than in our single-agent setup, leading to 1,2B frames instead of 200M.

Table 4: Raw agents scores after training Rainbow-IQN Ape-X with 10 actors or a single synchronized actor

| Raw score
Game | Multi-agent | Single agent |
|---|---|---|
| Asterix | 274,491 | 28,015 |
| Ms Pacman | 9,901 | 6,090.74 |
| Space Invaders | 24,183 | 7,385.4 |

## C GLITCH AND BUG IN THE ALE

Inconsistent game behaviors and bugs were encountered while benchmarking Rainbow and Rainbow-IQN on all Atari games. The most damageable is the one described in the main article: games getting stuck forever even doing random actions. This is one of the main reasons why the *maximum stuck length* parameter is introduced.

Another issue is the *buffer rollover*: the emulator sends a reward of -1M when reaching 1M, effectively making the agent goes to 0 score over and over. For example, for our first implementation of Rainbow on Asterix, the scores were going up to 1M, then suddenly collapsing to random values between 0 and 1M. However, the trained agent was in fact playing almost perfectly and was indeed resolving the game many times before dying. This can also be observed in the reported score of Asterix by both Ape-X (Horgan et al., 2018) and Rainbow (Hessel et al., 2018): the score goes up to 1M and then varies randomly. This is an issue to compare agent, because a weaker agent could actually be reported with a higher score. We found this kind of *buffer rollover* bug in 2 others games: Video Pinball and Defender. To detect this in potential other games, we advocate to keep track of really high negative rewards. Indeed on the 61 games evaluated, there are no game on which there is reward inferior to -1000. And if it happens, most probably this is a buffer rollover and this reward should be ignored.

Additionally, on many games (such as Breakout for example), a specific key must be pressed to start the game (most of the time the Fire button). This means that agent can easily get stuck for long time because it does not press the key. This impacts the stability of the training because the replay memory is filled with useless transitions. We argue that this problem is exacerbated by not finishing episode as loss of life. Indeed there are many games where a specific key must be pressed, but only after losing a life to continue the game. Moreover this is probably harder to learn with the whole action set available, because the number of actions to iterate on is higher than with the minimal useful action set. This is definitely not a bug, and a general agent should learn to press fire to restart or start game.

# D    DETAILED EXPERIMENTAL FIGURES

In this section, we provide more detailed versions of the figures in the main article. The structure of this section follows the one of Section 5 of the main article.

As a reminder, all *normalized world record baseline* scores $s$ are reported according to the following equation, where we note $r$ the score of a random agent, $w$ the score of the world record, and $a$ the score of the agent to be evaluated:

$$s = \frac{a - r}{|w - r|} \tag{1}$$

## D.1    RAINBOW EVALUATION

Figure 7 illustrates in more details the difference between the reported original performance of (Hessel et al., 2018) (reported in the world record baseline), and the one obtained when applying the recommendations of (Machado et al., 2018). In particular, the number of failing games is much lower for the original implementation. Figure 8 gives the breakdown for each game of the ALE.

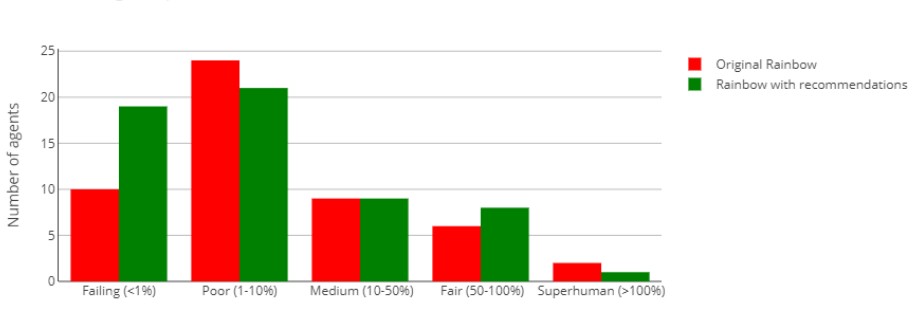

Figure 7: Agents performance comparison for the original Rainbow (Hessel et al., 2018) versus Rainbow trained with (Machado et al., 2018) guidelines (30 minutes evaluation time to align with original conditions)

## D.2    RAINBOW-IQN: EVALUATION AND COMPARISON

**Influence of maximum episode length**    Figure 9 details the influence of evaluation time over the performance range of the agents. As expected and discussed in the main article, evaluation time has a strong impact on the normalized performance of the agents. In particular, no agent reaches superhuman performance before 30 minutes evaluation. More agents reach superhuman performance when the evaluation time is not capped (in particular the ones that never stop playing, see next paragraph).

**Comparison of Rainbow and Rainbow-IQN**    Figure 10 details the difference in performance between Rainbow and Rainbow-IQN on SABER conditions, at 200M training frames. Note that superhuman, never ending scores are artificially capped at 200% of the baseline. The most drastic difference is found on the game *asteroids*, which goes from failing to superhuman performance.

Some failing games are still significantly improved: for example, *space invaders* is increased of roughly a factor of 3. To highlight these improvements, we compare Rainbow-IQN to Rainbow by using a normalized baseline similar to the world record baseline, but using Rainbow scores as a reference. So if we note $r$ the score of a random agent, $R$ the score of a Rainbow agent and $I$ the score of a Rainbow-IQN agent, then the normalized score $s$ is:

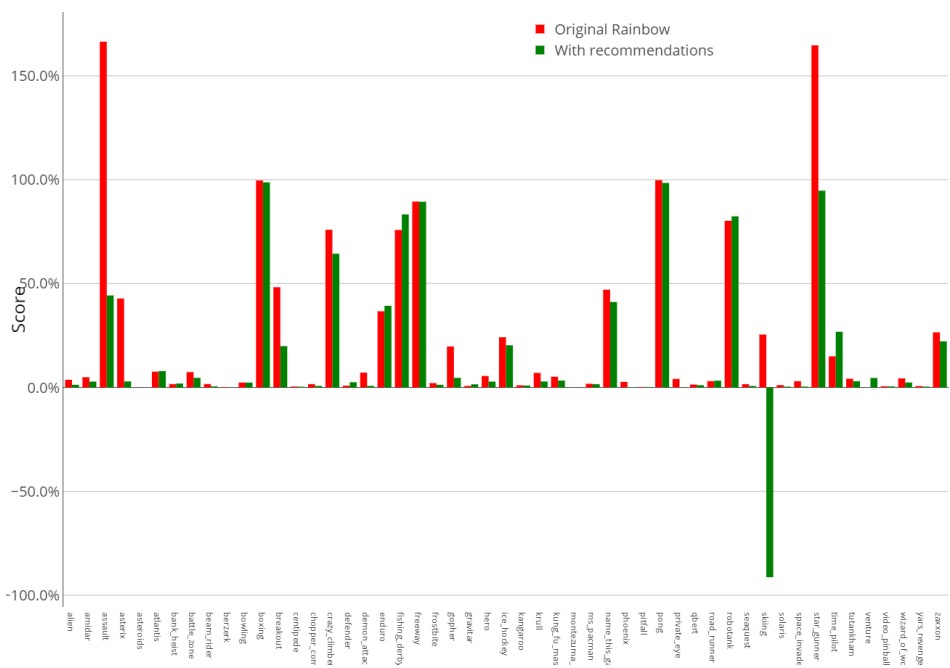

Figure 8: Performance comparison per game between the original Rainbow (Hessel et al., 2018) versus Rainbow trained with (Machado et al., 2018) guidelines (30 minutes evaluation time to align with original conditions)

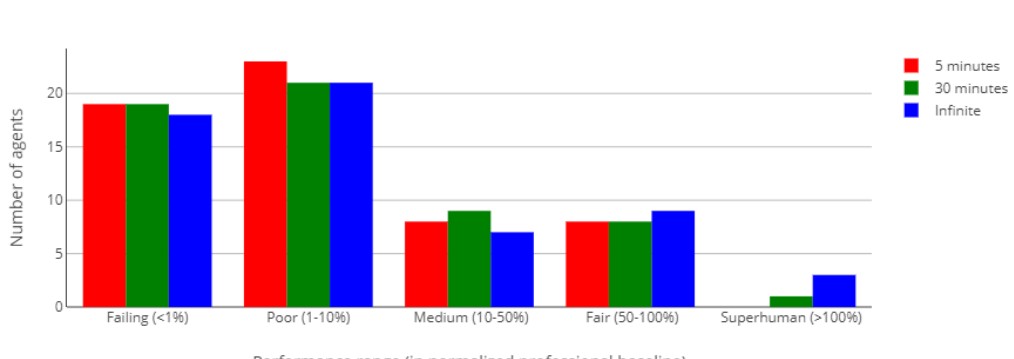

Figure 9: Evolution of agents performance classification with evaluation time: Rainbow-IQN, 200M training frames, evaluation time ranging from 5min to SABER conditions

$$s = \frac{I - r}{|R - r|} \qquad (2)$$

Note that we use the absolute value because in the game Skiing, the Rainbow agent is worse than the random agent. The details per game can be found in Figure 11. Note that games that are already superhuman in Rainbow are skipped, and that the Asteroids games, which is failing in Rainbow, becomes superhuman and is skipped in the figure for visualization purposes.

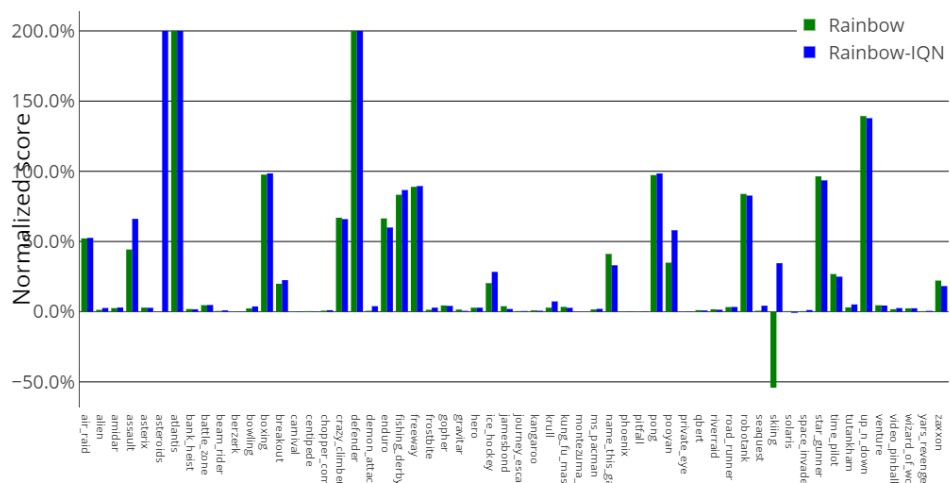

Figure 10: Performance comparison per game between Rainbow and Rainbow-IQN on SABER conditions (200M training frames)

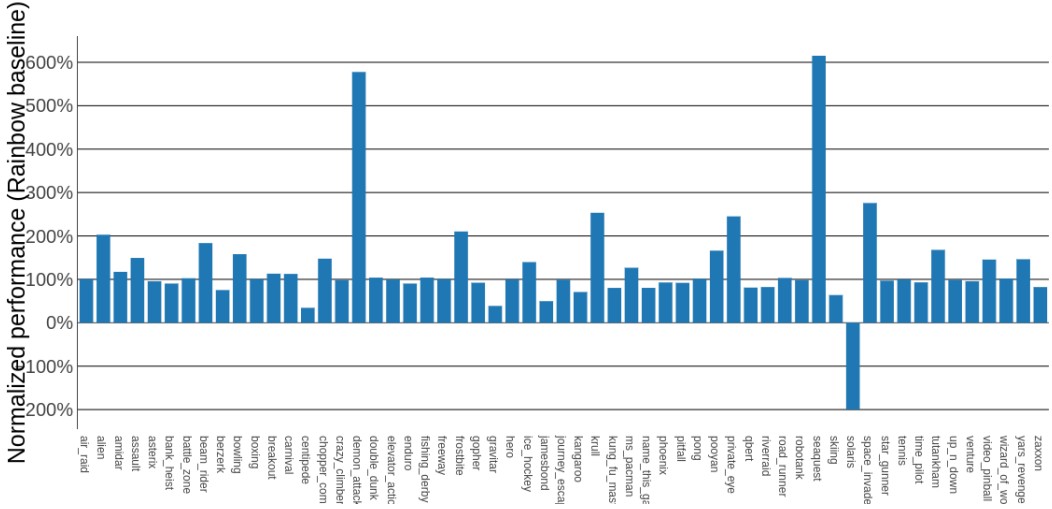

Figure 11: Rainbow-IQN normalized with regards to a Rainbow baseline for each game

### D.3 STABILITY OF BOTH RAINBOW AND RAINBOW-IQN

The 14 games on which we ran 5 trials for both Rainbow and Rainbow-IQN are: Asteroids, Centipede, Demon Attack, Frostbite, Gravitar, Jamesbond, Krull, Kung Fu Master, Ms Pacman, Private Eye, Seaquest, Up N Down, Yars Revenge and Zaxxon.

## E  RAW SCORES

For verification purposes, we provide tables containing all relevant agent scores used to build the figures from the principal article.

**Baseline scores**  Table 5 contains all raw game scores for ALE games, both for the previous human baseline (Mnih et al., 2015) and the new proposed world record baseline from TwinGalaxies. Note that some of the scores are missing for some games (marked as NA). For the world record baseline scores, some of them were extrapolated from the reported world record and are marked with a ∗. Indeed, some world records report the play time or other metrics (e.g. the distance travelled for

*Enduro*) instead of the raw score of the game. Note that all agents are trained and reported on all games of the ALE, even if the world record baseline is computed for 58 games.

**SABER raw scores for Rainbow-IQN**    Table 6 contains all raw agents scores for ALE games for Rainbow-IQN. A few of these games (Atlantis and Defender and Asteroids for Rainbow-IQN) successfully keep playing with a positive score increase after 100 hours, so their raw scores are infinite. They are marked as *infinite gameplay* in the table, and capped at 200% of the world record baseline for the mean computation.

**Evolution of scores with time**    Table 7 compares agents scores with increasing evaluation times for Rainbow and Rainbow-IQN, at 200M training frames.

**Evolution of scores with training frames**    Table 8 (resp. Table 9) contains all raw agents scores for ALE games for Rainbow-IQN, with an evaluation time of 5 minutes (resp. 30 minutes), after 10M, 50M, 100M and finally 200M training frames.

| Game Name | Random | Agent Category (Mnih et al., 2015) | World Record |
|---|---|---|---|
| air raid | 579.25 | NA | 23050.0 |
| alien | 211.9 | 7127.7 | 251916.0 |
| amidar | 2.34 | 1719.5 | 104159.0 |
| assault | 283.5 | 742.0 | 8647.0 |
| asterix | 268.5 | 8503.3 | 1000000.0 |
| asteroids | 1008.6 | 47388.7 | 10506650.0 |
| atlantis | 22188.0 | 29028.1 | 10604840.0 |
| bank heist | 14.0 | 753.1 | 82058.0 |
| battle zone | 3000.0 | 37187.5 | 801000.0 |
| beam rider | 414.32 | 16926.5 | 999999.0 |
| berzerk | 165.6 | 2630.4 | 1057940.0 |
| bowling | 23.48 | 160.7 | 300.0 |
| boxing | -0.69 | 12.1 | 100.0* |
| breakout | 1.5 | 30.5 | 864.0 |
| carnival | 700.8 | NA | 2541440.0 |
| centipede | 2064.77 | 12017.0 | 1301709.0 |
| chopper command | 794.0 | 7387.8 | 999999.0 |
| crazy climber | 8043.0 | 35829.4 | 219900.0 |
| defender | 4142.0 | 18688.9 | 6010500.0 |
| demon attack | 162.25 | 1971.0 | 1556345.0 |
| double dunk | -18.14 | -16.4 | NA |
| elevator action | 4387.0 | NA | NA |
| enduro | 0.01 | 860.5 | 9500.0* |
| fishing derby | -93.06 | -38.7 | 71.0 |
| freeway | 0.01 | 29.6 | 38.0 |
| frostbite | 73.2 | 4334.7 | 454830.0 |
| gopher | 364.0 | 2412.5 | 355040.0 |
| gravitar | 226.5 | 3351.4 | 162850.0 |
| hero | 551.0 | 30826.4 | 1000000.0 |
| ice hockey | -10.03 | 0.9 | 36.0 |
| jamesbond | 27.0 | 302.8 | 45550.0 |
| journey escape | -19977.0 | NA | 4317804.0 |
| kangaroo | 54.0 | 3035.0 | 1424600.0 |
| krull | 1566.59 | 2665.5 | 104100.0 |
| kung fu master | 451.0 | 22736.3 | 1000000.0 |
| montezuma revenge | 0.0 | 4753.3 | 1219200.0 |
| ms pacman | 242.6 | 6951.6 | 290090.0 |
| name this game | 2404.9 | 8049.0 | 25220.0 |
| phoenix | 757.2 | 7242.6 | 4014440.0 |
| pitfall | -265.0 | 6463.7 | 114000.0 |
| pong | -20.34 | 14.6 | 21.0* |
| pooyan | 371.2 | NA | 13025.0 |
| private eye | 34.49 | 69571.3 | 101800.0 |
| qbert | 188.75 | 13455.0 | 2400000.0 |
| riverraid | 1575.4 | 17118.0 | 1000000.0 |
| road runner | 7.0 | 7845.0 | 2038100.0 |
| robotank | 2.24 | 11.9 | 76.0 |
| seaquest | 88.2 | 42054.7 | 999999.0 |
| skiing | -16267.91 | -4336.9 | -3272.0* |
| solaris | 2346.6 | 12326.7 | 111420.0 |
| space invaders | 136.15 | 1668.7 | 621535.0 |
| star gunner | 631.0 | 10250.0 | 77400.0 |
| tennis | -23.92 | -8.3 | NA |
| time pilot | 3682.0 | 5229.2 | 65300.0 |
| tutankham | 15.56 | 167.6 | 5384.0 |
| up n down | 604.7 | 11693.2 | 82840.0 |
| venture | 0.0 | 1187.5 | 38900.0 |
| video pinball | 15720.98 | 17667.9 | 89218328.0 |
| wizard of wor | 534.0 | 4756.5 | 395300.0 |
| yars revenge | 3271.42 | 54576.9 | 15000105.0 |
| zaxxon | 8.0 | 9173.3 | 83700.0 |

Table 5: Raw scores for ALE games, for a random agent, the beginner baseline and the world records. * indicates games on which score has been extrapolated from the reported world record (the time of visit was 25 July 2019).                18

| Game name | Training frames | | | |
|---|---|---|---|---|
| | 10M | 50M | 100M | 200M |
| air raid | 7765.25 | 11690.0 | 13434.25 | 12289.75 |
| alien | 2740.6 | 1878.1 | 5223.0 | 7046.4 |
| amidar | 347.13 | 1554.84 | 2129.27 | 3092.05 |
| assault | 966.87 | 2783.49 | 4443.03 | 6372.7 |
| asterix | 3467.0 | 9280.0 | 16344.5 | 28015.0 |
| asteroids | 1194.16 (98.25) | 3261.88 (2602.48) | Infinite gameplay | Infinite gameplay |
| atlantis | Infinite gameplay | Infinite gameplay | Infinite gameplay | Infinite gameplay |
| bank heist | 756.4 | 1325.3 | 1402.2 | 1412.4 |
| battle zone | 33000.0 | 36730.0 | 33480.0 | 44410.0 |
| beam rider | 11510.78 | 11900.7 | 10042.74 | 9826.62 |
| berzerk | 546.7 | 697.0 | 640.2 | 892.9 |
| bowling | 29.64 | 30.0 | 29.86 | 29.92 |
| boxing | 92.71 | 98.62 | 98.92 | 98.7 |
| breakout | 53.77 | 121.83 | 132.56 | 175.47 |
| carnival | 5148.7 | 4824.1 | 4851.3 | 4566.3 |
| centipede | 2241.70 (251.56) | 4099.89 (405.19) | 4720.54 (626.31) | 5260.96 (920.10) |
| chopper command | 3018.0 | 6523.0 | 9053.0 | 11405.0 |
| crazy climber | 86310.0 | 118038.0 | 133114.0 | 144437.0 |
| defender | Infinite gameplay | Infinite gameplay | Infinite gameplay | Infinite gameplay |
| demon attack | 3433.13 (656.97) | 6616.96 (2949.90) | 8267.82 (3065.27) | 24599.31 (17441.86) |
| double dunk | -5.54 | 0.3 | 1.52 | 1.3 |
| elevator action | 2.0 | 0.0 | 43490.0 | 77010.0 |
| enduro | 1380.05 | 3867.42 | 5014.49 | 5146.73 |
| fishing derby | 22.11 | 34.82 | 48.11 | 49.08 |
| freeway | 32.65 | 33.9 | 33.95 | 33.96 |
| frostbite | 4351.54 (1456.01) | 9135.10 (1611.78) | 9768.28 (1742.88) | 10002.78 (1752.75) |
| gopher | 4798.4 | 15629.8 | 14136.0 | 15797.6 |
| gravitar | 283.70 (56.37) | 1258.90 (228.31) | 1725.90 (471.00) | 1973.60 (614.80) |
| hero | 13728.55 | 27450.65 | 28759.85 | 28957.4 |
| ice hockey | -2.43 | 1.8 | -0.72 | -0.07 |
| jamesbond | 445.70 (33.50) | 609.70 (46.97) | 605.00 (37.52) | 870.80 (171.30) |
| journey escape | -2096.0 | -1116.0 | -780.0 | -736.0 |
| kangaroo | 1740.0 | 4416.0 | 7088.0 | 9567.0 |
| krull | 6780.10 (467.12) | 8804.04 (97.77) | 9132.15 (207.84) | 9409.73 (98.14) |
| kung fu master | 24102.80 (6513.61) | 27867.00 (5783.19) | 28905.80 (6570.13) | 33312.00 (4119.74) |
| montezuma revenge | 0.0 | 0.0 | 0.0 | 0.0 |
| ms pacman | 2276.30 (144.50) | 5058.96 (602.27) | 5871.52 (454.28) | 6755.47 (555.18) |
| name this game | 10702.2 | 9702.9 | 10094.5 | 9946.4 |
| phoenix | 4586.7 | 5145.4 | 5370.6 | 5505.8 |
| pitfall | 0.0 | -3.95 | -2.74 | -21.34 |
| pong | 6.76 | 19.77 | 19.86 | 20.35 |
| pooyan | 4989.7 | 6334.05 | 6339.2 | 6776.7 |
| private eye | 99.40 (1.20) | 144.64 (46.57) | 173.02 (39.13) | 164.31 (42.75) |
| qbert | 4343.75 | 14809.5 | 16812.5 | 18736.25 |
| riverraid | 3955.9 | 15068.6 | 15891.3 | 15655.7 |
| road runner | 32737.0 | 51383.0 | 54599.0 | 67962.0 |
| robotank | 30.66 | 53.55 | 57.18 | 62.68 |
| seaquest | 3077.86 (131.08) | 21853.50 (4243.86) | 29694.50 (6157.97) | 46735.26 (10631.30) |
| skiing | -27031.73 | -20930.88 | -21053.79 | -12295.78 |
| solaris | 2027.2 | 2770.2 | 2205.2 | 1495.4 |
| space invaders | 695.15 | 1748.45 | 3365.2 | 10110.4 |
| star gunner | 13345.0 | 52961.0 | 59574.0 | 72441.0 |
| tennis | -3.19 | -0.02 | -0.07 | -0.03 |
| time pilot | 6501.0 | 11598.0 | 13550.0 | 19050.0 |
| tutankham | 128.7 | 177.96 | 284.42 | 288.41 |
| up n down | 18544.78 (3272.37) | 44569.10 (12243.70) | 56722.56 (9966.49) | 110907.70 (10256.62) |
| venture | 0.0 | 1046.0 | 1486.0 | 1679.0 |
| video pinball | 40107.82 | 1784770.52 | 3008620.51 | 1254569.69 |
| wizard of wor | 4133.0 | 7441.0 | 7466.0 | 9369.0 |
| yars revenge | 11077.61 (1366.42) | 72860.33 (7560.21) | 84238.64 (7721.16) | 93144.71 (5251.19) |
| zaxxon | 8319.00 (557.20) | 12494.80 (282.63) | 14077.60 (917.33) | 13913.40 (585.68) |

Table 6: Raw scores for ALE game agents trained with Rainbow-IQN on SABER at 10M, 50M, 100M and 200M training frames. For the 14 games ran on 5 seeds, we also show the standard deviation.

| Game name | Rainbow 5 minutes | Rainbow 30 minutes | Rainbow SABER | Rainbow-IQN 5 minutes | Rainbow-IQN 30 minutes | Rainbow-IQN SABER |
|---|---|---|---|---|---|---|
| air raid | 10549 | 12308.25 | 12308.25 | 11107.25 | 12289.75 | 12289.75 |
| alien | 3458.5 | 3458.5 | 3458.5 | 7046.4 | 7046.4 | 7046.4 |
| amidar | 2835.53 | 2952.43 | 2952.43 | 2601.82 | 3092.05 | 3092.05 |
| assault | 3779.98 | 3986.1 | 3986.1 | 5178.41 | 6372.7 | 6372.7 |
| asterix | 29269 | 29269 | 29269 | 28015.0 | 28015.0 | 28015 |
| asteroids | 1716.90 (238) | 1716.90 (238) | 1716.90 (238) | 30838.86 | 159426.4 | Infinite gameplay |
| atlantis | 129392 | 858765 | Infinite gameplay | 130475.0 | 839433.0 | Infinite gameplay |
| bank heist | 1563.2 | 1563.2 | 1563.2 | 1412.4 | 1412.4 | 1412.4 |
| battle zone | 45610 | 45610 | 45610 | 44410.0 | 44410.0 | 44410 |
| beam rider | 5437.14 | 5542.22 | 5542.22 | 8165.14 | 9826.62 | 9826.62 |
| berzerk | 1049.3 | 1049.3 | 1049.3 | 888.0 | 892.9 | 892.9 |
| bowling | 29.92 | 29.92 | 29.92 | 29.92 | 29.92 | 29.92 |
| boxing | 98.7 | 98.7 | 98.7 | 98.7 | 98.7 | 98.7 |
| breakout | 173.01 | 173.01 | 173.01 | 175.39 | 175.47 | 175.47 |
| carnival | 4163.5 | 4163.5 | 4163.5 | 4566.3 | 4566.3 | 4566.3 |
| centipede | 7267.82 (265) | 7267.82 (265) | 7267.82 (265) | 5260.96 | 5260.96 | 5260.96 |
| chopper command | 7973 | 7973 | 7973 | 11405.0 | 11405.0 | 11405 |
| crazy climber | 133756 | 144373 | 144373 | 137299.0 | 144437.0 | 144437 |
| defender | 18524.71 | 30976.24 | Infinite gameplay | 19004.03 | 24926.15 | Infinite gameplay |
| demon attack | 10234.20 (415) | 14617.11 (2215) | 14617.11 (2215) | 10294.51 | 24596.37 | 24599.31 |
| double dunk | 0 | 0 | 0 | 1.1 | 1.3 | 1.3 |
| elevator action | 13421 | 85499 | 85499 | 12455.0 | 77010.0 | 77010 |
| enduro | 369.87 | 2332.63 | 6044.36 | 373.3 | 2316.67 | 5146.73 |
| fishing derby | 43.57 | 43.57 | 43.57 | 49.08 | 49.08 | 49.08 |
| freeway | 33.96 | 33.96 | 33.96 | 33.96 | 33.96 | 33.96 |
| frostbite | 7075.14 (656) | 7075.14 (656) | 7075.14 (656) | 10002.78 | 10002.78 | 10002.78 |
| gopher | 12405 | 16736.4 | 16736.4 | 11724.8 | 15797.6 | 15797.6 |
| gravitar | 2647.50 (398) | 2647.50 (398) | 2647.50 (398) | 1973.6 | 1973.6 | 1973.6 |
| hero | 28911.15 | 28911.15 | 28911.15 | 28957.4 | 28957.4 | 28957.4 |
| ice hockey | -0.69 | -0.69 | -0.69 | -0.07 | -0.07 | -0.07 |
| jamesbond | 1421.00 (502) | 1434.00 (509) | 1434.00 (509) | 870.8 | 870.8 | 870.8 |
| journey escape | -645 | -645 | -645 | -736.0 | -736.0 | -736 |
| kangaroo | 13242 | 13242 | 13242 | 9567.0 | 9567.0 | 9567 |
| krull | 4697.19 (273) | 4697.19 (273) | 4697.19 (273) | 9409.73 | 9409.73 | 9409.73 |
| kung fu master | 32265.20 (6476) | 32692.80 (6790) | 32692.80 (6790) | 32934.8 | 33312.0 | 33312 |
| montezuma revenge | 0 | 0 | 0 | 0.0 | 0.0 | 0 |
| ms pacman | 4738.30 (265) | 4738.30 (265) | 4738.30 (265) | 6755.47 | 6755.47 | 6755.47 |
| name this game | 8187.4 | 11787.7 | 11787.7 | 7579.8 | 9946.4 | 9946.4 |
| phoenix | 5943.9 | 5943.9 | 5943.9 | 5505.8 | 5505.8 | 5505.8 |
| pitfall | 0 | 0 | 0 | -11.11 | -21.34 | -21.34 |
| pong | 20.35 | 20.35 | 20.35 | 20.35 | 20.35 | 20.35 |
| pooyan | 4766.3 | 4788.5 | 4788.5 | 6466.6 | 6776.7 | 6776.7 |
| private eye | 100.00 (0) | 100.00 (0) | 100.00 (0) | 164.31 | 164.31 | 164.31 |
| qbert | 26116 | 26171.75 | 26171.75 | 18736.25 | 18736.25 | 18736.25 |
| riverraid | 18456 | 18456 | 18456 | 15655.7 | 15655.7 | 15655.7 |
| road runner | 66593 | 66593 | 66593 | 67962.0 | 67962.0 | 67962 |
| robotank | 52.34 | 62.99 | 62.99 | 51.35 | 62.68 | 62.68 |
| seaquest | 12281.82 (7018) | 20670.40 (17377) | 20670.40 (17377) | 28554.0 | 46735.26 | 46735.26 |
| skiing | -28105.83 | -28134.23 | -28134.23 | -12294.58 | -12295.78 | -12295.78 |
| solaris | 2299.4 | 2779.4 | 2779.4 | 819.0 | 1495.4 | 1495.4 |
| space invaders | 2764.55 | 2764.55 | 2764.55 | 4718.2 | 10110.4 | 10110.4 |
| star gunner | 72944 | 73331 | 73331 | 71705.0 | 72441.0 | 72441 |
| tennis | 0 | 0 | 0 | -0.03 | -0.03 | -0.03 |
| time pilot | 20198 | 20198 | 20198 | 19050.0 | 19050.0 | 19050 |
| tutankham | 177.17 | 177.42 | 177.42 | 288.41 | 288.41 | 288.41 |
| up n down | 52599 (4454) | 105213 (23843) | 105213 (23843) | 56646.0 | 110655.76 | 110907.7 |
| venture | 1781 | 1781 | 1781 | 1679.0 | 1679.0 | 1679 |
| video pinball | 96345.36 | 656571.52 | 2197677.95 | 76587.14 | 465419.66 | 1254569.69 |
| wizard of wor | 9913 | 9943 | 9943 | 9369.0 | 9369.0 | 9369 |
| yars revenge | 60913 (2342) | 60913 (2342) | 60913 (2342) | 93144.71 | 93144.71 | 93144.71 |
| zaxxon | 19017 (1228) | 19060 (1238) | 19060 (1238) | 13913.4 | 13913.4 | 13913.4 |

Table 7: Agent scores for Rainbow and Rainbow-IQN at 200M training frames, reported for 5min, 30min and SABER (no limit) evaluation time. Standard deviation are showed for Rainbow (for Rainbow-IQN it can be found on next tables). 20

| | Training frames | | | |
|---|---|---|---|---|
| Game name | 10M | 50M | 100M | 200M |
| air raid | 7549.0 | 9168.75 | 10272.75 | 11107.25 |
| alien | 2740.6 | 1878.1 | 5223.0 | 7046.4 |
| amidar | 347.13 | 1554.84 | 2129.27 | 2601.82 |
| assault | 966.87 | 2783.49 | 4103.89 | 5178.41 |
| asterix | 3467.0 | 9280.0 | 16344.5 | 28015.0 |
| asteroids | 1194.16 (98.25) | 3251.64 (2582.07) | 12261.36 (12251.18) | 30838.86 (5427.63) |
| atlantis | 101945.0 | 118844.0 | 125696.0 | 130475.0 |
| bank heist | 756.4 | 1325.3 | 1402.2 | 1412.4 |
| battle zone | 33000.0 | 36730.0 | 33480.0 | 44410.0 |
| beam rider | 6764.82 | 8554.82 | 7818.72 | 8165.14 |
| berzerk | 546.7 | 697.0 | 640.2 | 888.0 |
| bowling | 29.64 | 30.0 | 29.86 | 29.92 |
| boxing | 92.71 | 98.62 | 98.92 | 98.7 |
| breakout | 53.77 | 121.83 | 132.56 | 175.39 |
| carnival | 5148.7 | 4824.1 | 4851.3 | 4566.3 |
| centipede | 2241.70 (251.56) | 4099.89 (405.19) | 4720.54 (626.31) | 5260.96 (920.10) |
| chopper command | 3018.0 | 6523.0 | 9053.0 | 11405.0 |
| crazy climber | 86085.0 | 117582.0 | 130559.0 | 137299.0 |
| defender | 36353.98 | 19608.36 | 18915.17 | 19004.03 |
| demon attack | 3383.66 (648.57) | 5833.77 (1542.65) | 7161.13 (1364.32) | 10294.51 (1868.54) |
| double dunk | -5.24 | 0.3 | 1.52 | 1.1 |
| elevator action | 2.0 | 0.0 | 7360.0 | 12455.0 |
| enduro | 340.68 | 379.34 | 382.91 | 373.3 |
| fishing derby | 22.11 | 34.82 | 48.11 | 49.08 |
| freeway | 32.65 | 33.9 | 33.95 | 33.96 |
| frostbite | 4351.54 (1456.01) | 9135.10 (1611.78) | 9768.28 (1742.88) | 10002.78 (1752.75) |
| gopher | 4798.4 | 11561.0 | 10944.4 | 11724.8 |
| gravitar | 283.70 (56.37) | 1258.90 (228.31) | 1725.90 (471.00) | 1973.60 (614.80) |
| hero | 13728.55 | 27450.65 | 28759.85 | 28957.4 |
| ice hockey | -2.43 | 1.8 | -0.72 | -0.07 |
| jamesbond | 445.70 (33.50) | 609.70 (46.97) | 605.00 (37.52) | 870.80 (171.30) |
| journey escape | -2096.0 | -1116.0 | -780.0 | -736.0 |
| kangaroo | 1740.0 | 4416.0 | 7088.0 | 9567.0 |
| krull | 6780.10 (467.12) | 8804.04 (97.77) | 9132.15 (207.84) | 9409.73 (98.14) |
| kung fu master | 23970.80 (6513.13) | 27701.20 (5814.09) | 28708.80 (6580.12) | 32934.80 (4170.04) |
| montezuma revenge | 0.0 | 0.0 | 0.0 | 0.0 |
| ms pacman | 2276.30 (144.50) | 5058.96 (602.27) | 5871.52 (454.28) | 6755.47 (555.18) |
| name this game | 8212.4 | 7790.3 | 7754.6 | 7579.8 |
| phoenix | 4586.7 | 5145.4 | 5370.6 | 5505.8 |
| pitfall | 0.0 | -3.95 | -2.58 | -11.11 |
| pong | 6.29 | 19.77 | 19.86 | 20.35 |
| pooyan | 4956.6 | 6233.55 | 6183.95 | 6466.6 |
| private eye | 99.40 (1.20) | 144.64 (46.57) | 173.02 (39.13) | 164.31 (42.75) |
| qbert | 4343.75 | 14809.5 | 16812.5 | 18736.25 |
| riverraid | 3955.9 | 15068.6 | 15891.3 | 15655.7 |
| road runner | 32737.0 | 51383.0 | 54426.0 | 67962.0 |
| robotank | 25.0 | 42.14 | 45.56 | 51.35 |
| seaquest | 3077.86 (131.08) | 18200.66 (2114.98) | 21750.36 (1891.98) | 28554.00 (3617.42) |
| skiing | -27012.53 | -20923.28 | -21046.99 | -12294.58 |
| solaris | 1210.6 | 1552.4 | 1338.0 | 819.0 |
| space invaders | 695.15 | 1748.45 | 3347.25 | 4718.2 |
| star gunner | 13345.0 | 52961.0 | 59572.0 | 71705.0 |
| tennis | -3.19 | -0.02 | -0.04 | -0.03 |
| time pilot | 6501.0 | 11598.0 | 13550.0 | 19050.0 |
| tutankham | 128.7 | 177.71 | 284.42 | 288.41 |
| up n down | 14722.22 (2551.46) | 35663.92 (7724.72) | 42380.48 (4978.69) | 56646.00 (2541.90) |
| venture | 0.0 | 1046.0 | 1486.0 | 1679.0 |
| video pinball | 29524.06 | 122029.58 | 79508.52 | 76587.14 |
| wizard of wor | 4133.0 | 7441.0 | 7466.0 | 9369.0 |
| yars revenge | 11077.61 (1366.42) | 72860.33 (7560.21) | 84238.64 (7721.16) | 93144.71 (5251.19) |
| zaxxon | 8319.00 (557.20) | 12494.80 (282.63) | 14073.20 (910.64) | 13913.40 (585.68) |

Table 8: Raw scores for ALE game agents trained for Rainbow-IQN at 10M, 50M, 100M and 200M training frames for 5 minutes evaluation. For the 14 games ran on 5 seeds, we also show the standard deviation.

| Game name | Training frames | | | |
| --- | --- | --- | --- | --- |
| | 10M | 50M | 100M | 200M |
| air raid | 7765.25 | 11690.0 | 13434.25 | 12289.75 |
| alien | 2740.6 | 1878.1 | 5223.0 | 7046.4 |
| amidar | 347.13 | 1554.84 | 2129.27 | 3092.05 |
| assault | 966.87 | 2783.49 | 4443.03 | 6372.7 |
| asterix | 3467.0 | 9280.0 | 16344.5 | 28015.0 |
| asteroids | 1194.16 (98.25) | 3261.88 (2602.48) | 48027.06 (56599.58) | 159426.40 (56987.42) |
| atlantis | 261697.0 | 788006.0 | 817118.0 | 839433.0 |
| bank heist | 756.4 | 1325.3 | 1402.2 | 1412.4 |
| battle zone | 33000.0 | 36730.0 | 33480.0 | 44410.0 |
| beam rider | 11510.78 | 11900.7 | 10042.74 | 9826.62 |
| berzerk | 546.7 | 697.0 | 640.2 | 892.9 |
| bowling | 29.64 | 30.0 | 29.86 | 29.92 |
| boxing | 92.71 | 98.62 | 98.92 | 98.7 |
| breakout | 53.77 | 121.83 | 132.56 | 175.47 |
| carnival | 5148.7 | 4824.1 | 4851.3 | 4566.3 |
| centipede | 2241.70 (251.56) | 4099.89 (405.19) | 4720.54 (626.31) | 5260.96 (920.10) |
| chopper command | 3018.0 | 6523.0 | 9053.0 | 11405.0 |
| crazy climber | 86310.0 | 118038.0 | 133114.0 | 144437.0 |
| defender | 49409.81 | 35899.7 | 24663.27 | 24926.15 |
| demon attack | 3433.13 (656.97) | 6616.96 (2949.90) | 8267.82 (3065.27) | 24596.37 (17442.46) |
| double dunk | -5.54 | 0.3 | 1.52 | 1.3 |
| elevator action | 2.0 | 0.0 | 43490.0 | 77010.0 |
| enduro | 1378.3 | 2242.11 | 2307.42 | 2316.67 |
| fishing derby | 22.11 | 34.82 | 48.11 | 49.08 |
| freeway | 32.65 | 33.9 | 33.95 | 33.96 |
| frostbite | 4351.54 (1456.01) | 9135.10 (1611.78) | 9768.28 (1742.88) | 10002.78 (1752.75) |
| gopher | 4798.4 | 15629.8 | 14136.0 | 15797.6 |
| gravitar | 283.70 (56.37) | 1258.90 (228.31) | 1725.90 (471.00) | 1973.60 (614.80) |
| hero | 13728.55 | 27450.65 | 28759.85 | 28957.4 |
| ice hockey | -2.43 | 1.8 | -0.72 | -0.07 |
| jamesbond | 445.70 (33.50) | 609.70 (46.97) | 605.00 (37.52) | 870.80 (171.30) |
| journey escape | -2096.0 | -1116.0 | -780.0 | -736.0 |
| kangaroo | 1740.0 | 4416.0 | 7088.0 | 9567.0 |
| krull | 6780.10 (467.12) | 8804.04 (97.77) | 9132.15 (207.84) | 9409.73 (98.14) |
| kung fu master | 24102.80 (6513.61) | 27867.00 (5783.19) | 28905.80 (6570.13) | 33312.00 (4119.74) |
| montezuma revenge | 0.0 | 0.0 | 0.0 | 0.0 |
| ms pacman | 2276.30 (144.50) | 5058.96 (602.27) | 5871.52 (454.28) | 6755.47 (555.18) |
| name this game | 10702.2 | 9702.9 | 10094.5 | 9946.4 |
| phoenix | 4586.7 | 5145.4 | 5370.6 | 5505.8 |
| pitfall | 0.0 | -3.95 | -2.74 | -21.34 |
| pong | 6.76 | 19.77 | 19.86 | 20.35 |
| pooyan | 4989.7 | 6334.05 | 6339.2 | 6776.7 |
| private eye | 99.40 (1.20) | 144.64 (46.57) | 173.02 (39.13) | 164.31 (42.75) |
| qbert | 4343.75 | 14809.5 | 16812.5 | 18736.25 |
| riverraid | 3955.9 | 15068.6 | 15891.3 | 15655.7 |
| road runner | 32737.0 | 51383.0 | 54599.0 | 67962.0 |
| robotank | 30.66 | 53.55 | 57.18 | 62.68 |
| seaquest | 3077.86 (131.08) | 21853.50 (4243.86) | 29694.50 (6157.97) | 46735.26 (10631.30) |
| skiing | -27031.73 | -20930.88 | -21053.79 | -12295.78 |
| solaris | 2027.2 | 2770.2 | 2205.2 | 1495.4 |
| space invaders | 695.15 | 1748.45 | 3365.2 | 10110.4 |
| star gunner | 13345.0 | 52961.0 | 59574.0 | 72441.0 |
| tennis | -3.19 | -0.02 | -0.07 | -0.03 |
| time pilot | 6501.0 | 11598.0 | 13550.0 | 19050.0 |
| tutankham | 128.7 | 177.96 | 284.42 | 288.41 |
| up n down | 18516.40 (3286.13) | 44569.10 (12243.70) | 56722.56 (9966.49) | 110655.76 (10325.07) |
| venture | 0.0 | 1046.0 | 1486.0 | 1679.0 |
| video pinball | 40107.82 | 798642.24 | 565903.18 | 465419.66 |
| wizard of wor | 4133.0 | 7441.0 | 7466.0 | 9369.0 |
| yars revenge | 11077.61 (1366.42) | 72860.33 (7560.21) | 84238.64 (7721.16) | 93144.71 (5251.19) |
| zaxxon | 8319.00 (557.20) | 12494.80 (282.63) | 14077.60 (917.33) | 13913.40 (585.68) |

Table 9: Raw scores for ALE game agents trained for Rainbow-IQN at 10M, 50M, 100M and 200M training frames for 30 minutes evaluation. For the 14 games ran on 5 seeds, we also show the standard deviation.

