# OpenReview forum: "Is Deep Reinforcement Learning Really Superhuman on Atari? Leveling the playing field"
_ICLR.cc/2020/Conference — Reject_

### Official Review · AnonReviewer1 · 2019-10-21
**Official Blind Review #1**

**Rating:** 3

**Review:**


This paper proposes a new way to benchmark DRL algorithms using the Atari environment which is twofold, one part is a set of emulator recommendations, the other part is what quantity we should consider as a "human reference". The paper also compares Rainbow and Rainbow-IQN, where the IQN improvement matches the proposed human normalized score improvment.

I'm not quite sure how to rate this paper, I have put weak-reject for now, as I don't strongly disagree with anything in the paper, but at the same time:
- the difference to Machados et al. is marginal, but is a bit surprising
- the Rainbow-IQN improvement is too incremental to be considered a significant contribution
- there are some interesting remarks on why Atari is _not_ necessarily a good environment, e.g. most of Section 6, but this clashes with the paper's premise that we should be using Atari.

In a way, this paper reads like an interesting technical review of Atari, but I don't think it provides enough new knowledge to be a conference paper.

Detailed comments:
- I find it a bit weird that the many weaknesses of Atari as a platform are presented at DRL being bad at Atari. The line between environment design and algorithm design can be blurry, but in Atari's case, the weird peculiarities of each game are known to make it an inconvenient benchmark.
- In the same vein, why is Atari+SABER better than other RL environments? This is rather crucial. We should only work on improving a benchmark if it is a useful benchmark, yet, we have many clues that Atari is not.
- The link to TwinGalaxies should be a proper reference with the time of visit, especially if humans break new records in the future.
- Why only compare Rainbow and a variant of Rainbow? I understand compute resources being a limitation, but at the same time, the reasoning behind having standardized testing is to be able to compare a wide variety of algorithms. This paper would be much stronger if it focused on a few representative games (e.g. one reflex game, one hard exploration game, etc.) and tested these games with a bunch of DRL algorithms. That ranking might reveal something very interesting.
- The paper is easy to read, but there are a few grammar mistakes here and there.

**Experience Assessment:**

I have published one or two papers in this area.

**Review Assessment: Checking Correctness Of Derivations And Theory:**

N/A

**Review Assessment: Checking Correctness Of Experiments:**

I assessed the sensibility of the experiments.

**Review Assessment: Thoroughness In Paper Reading:**

I read the paper thoroughly.

---

> ### Author Response · Authors · 2019-11-07
> **Answer to Reviewer #1**
>
> Thank you for your comments and feedback. We will try to answer to all your questions and remarks in the following.
>
> "the difference to Machados et al. is marginal, but is a bit surprising": We think the difference with Machado et al. is not marginal, because we expose a significant divergence source, the game evaluation time. Machado et al. used a 5 minutes maximum length, to be compared with most of other works using 30 minutes. Moreover as mentioned in the main paper, we think this parameter is a source of ambiguity and led to mistakes even on important published papers in the field.
>
> "there are some interesting remarks on why Atari is not necessarily a good environment, e.g. most of Section 6": The authors are in fact convinced that Atari is a good environment for general RL and AI. Indeed, the Section 6 gives some hypotheses on why the RL algorithms are that bad, not that the ALE environment is. In fact, those remarks are even an explanation of why Atari is indeed a good environment describing its remaining difficulties and variability. The authors actually think that on the 3 most commonly used benchmark for DRL: ALE, DeepMind Lab and MuJoCo, ALE is the one with the most variability. We argue that this variability comes from the fact that those games were designed for human players and human don't like to play games that are too similar. In conclusion, we think that ALE is still a really interesting environment for general AI because current DRL algorithms are far from solving most of Atari games.
>
> "The link to TwinGalaxies should be a proper reference with the time of visit": We described all the world records we used in the Supplementary Materials, this allows easy reproducibility and comparison and avoid the cumbersome process of checking world records one by one. As suggested, we have specifically included the time of visit for additional clarity in the revision we just submitted.
>
> "This paper would be much stronger if it focused on a few representative games (e.g. one reflex game, one hard exploration game, etc.) and tested these games with a bunch of DRL algorithms": We agree it would have been really interesting to evaluate different DRL algorithms but this was not possible regarding our computational resources. Regarding testing on few representative games, we actually think this can bias the results and not represent the generality of the algorithm tested. A concurrent submission at ICLR, https://openreview.net/forum?id=BJewlyStDr , actually supports our argument. Their abstract claims "the real pace of progress in exploration research for Atari 2600 games may have been obfuscated by good results on a single domain". Indeed they showed that most of advance for hard-exploration games were most of the time overfitting to Montezuma Revenge and not really leading to improvement on other hard-exploration games and often even led to lower performances on easy exploration Atari games.

---

> > ### Comment · AnonReviewer1 · 2019-11-13
> > **Additional Comments**
> >
> > Thank you for your answers.
> >
> > 1) As in the famous xkcd comic about making new standards, this paper proposes a new standard way of testing and reporting Atari results. While it is demonstrably better, it isn't clear to me that it will allow researchers to much better distinguish between RL algorithms.
> >
> > 2) The problem with ALE being "designed for human players" is exactly that. Humans have years of prior knowledge about objects, embodiment, physics, and many more. It's easier to learn from a general distribution and apply the solution to cartoon problems than to learn from cartoons and apply the solution to real world general problems. This is fairly well established by the success of sim2real. As such, I remain unconvinced that the ALE is the best currently available benchmark for general purpose RL algorithm.
> >
> > 4) Thank you for the link, I will take this into consideration in the future. I do think, though, that this is further evidence that the ALE isn't a great research platform. Researchers rarely have the resources to test all games simultaneously, and since transfer from game to game is very poor, papers tend to overfit to some environments (as the concurrent submission points out).

---

> > > ### Author Response · Authors · 2019-11-14
> > > **Additional answer to 1) and 2)**
> > >
> > > Thank you for the additional comments.
> > >
> > > 1) We would like to insist on the fact that the goal of our article is not only to provide yet another standard, but also to point out the imprecision of the existing comparisons. Typically, not specifying the evaluation time, or setting it to 5 min where scores can be saturated, can make comparison irrelevant. Another goal of SABER is also to provide a better estimate of where RL agents stand relatively to human players, and so to show that there is still a large margin of progression before considering the ALE environment as solved.
> > >
> > > 2) ALE is indeed a benchmark that requires a lot of prior knowledge, so solving it is effectively biased towards that problem. However, the question of how to integrate this knowledge in a generic RL algorithm remains an open and interesting question, and we feel the ALE is the ideal testing ground for it.

---

### Official Review · AnonReviewer3 · 2019-10-23
**Official Blind Review #3**

**Rating:** 3

**Review:**

The paper proposes an extension to the work of Machado et al. (2018) for standardizing training and evaluation procedures in the Arcade Learning Environment (ALE). It then introduces a collection of human world records for each Atari game to refute previous claims of superhuman performance, as well as recommend comparisons against these records. They proceed to evaluate Rainbow under their proposed evaluation procedures, as well as introduce a new algorithm, Rainbow-IQN, with similar evaluations made based on their proposal.

I'm proposing a weak rejection as I feel some of the arguments made in the paper aren't very strong. In particular, I'd like the authors to comment on the following:

1) The key difference between their evaluation benchmark and the recommendations in Machado et al. (2018) are that episodes should not have a time limit. The justification for this is that many algorithms might achieve practically optimal performance within this time limit, and so one wouldn't be able to compare algorithms on certain games within significance. They further emphasize that human high scores were achieved without limiting to 30 minutes of play. That said, several algorithms performing similarly within said limit can be instead interpreted as shifting emphasis toward comparing performance on the harder games. As the paper acknowledged, removing the maximum episode length ended up introducing more issues, such as the emulator never ending an episode (due to a supposed bug), as well as increasing the likelihood of the score overflowing. The paper suggested a trick of limiting how long an agent can go without receiving a reward, but it's unclear (1) if needing this fix is worth the proposed change, and (2) if the fix introduces additional game-specific nuances in evaluation; e.g., are there any situations where this can be detrimental to properly evaluating performance, or introduce biases based on a game's reward distribution?

2) The paper gathered a list of human world records for the Atari games in the ALE. In my opinion, this is very valuable for the literature in terms of addressing prior work misrepresenting the competency of an algorithm relative to what humans are capable of; a professional game tester is supposed to be representative of the average game player, who is typically tasked with optimizing fun, whereas speedrunners and scorerunners of games are tasked with optimizing a comparable objective to an RL agent. Beyond this though, I think an alternative conclusion would be to use this information in support of not comparing results to human scores, and to focus on comparisons between algorithms. A considerable number of the human world records have reached the maximum allowable score, over drastically variable gameplay times to achieve these scores, that it might still not be that fair a comparison. Have the authors considered this possibility?

3) Were any other algorithms evaluated on this benchmark, beyond Rainbow? While there are computational considerations, it seems odd for a benchmarking-focused paper to only evaluate one standard algorithm and slight modification of it.

Suggestions

1) The introduction of Rainbow-IQN in this paper feels a little random and out of place given the context created by the rest of the paper's contributions- I feel it might be more appropriate for a benchmarking paper to focus on a representative set of "standard" or relatively simple/trivial algorithms (Like Machado et al. (2018) did) to give a frame of reference for comparing novel ones.

**Experience Assessment:**

I have published one or two papers in this area.

**Review Assessment: Checking Correctness Of Derivations And Theory:**

N/A

**Review Assessment: Checking Correctness Of Experiments:**

I assessed the sensibility of the experiments.

**Review Assessment: Thoroughness In Paper Reading:**

I read the paper thoroughly.

---

> ### Author Response · Authors · 2019-11-07
> **Answer to Reviewer #3**
>
> Thank you for your comments and feedback. We will try to answer to all your questions and remarks in the following.
>
> 1) It is true that removing the maximum length time introduced some issues but we think those issues are easily solved with our $\textit{maximum stuck time}$. On the other hand, the fact that 2 main papers (PPO and C51) did a mistake with this parameter is a good indicator of how much such limitation is a source of error and ambiguity. Moreover Machado et al. used only 5 minutes whereas some other papers used 30 minutes, which is an example of a comparison issue.
> Concerning the possible biases based on game's reward distribution, we actually tested both 5 and 30 minutes as maximum stuck time. It led to the exact same results: being stuck 5 min always meant being stuck over 30 min. Moreover Machado et al. used 5 minutes as their max episode length, implying that 5 min is a significant unit of time for an ALE game. This is comforted when considering those games were designed with the patience of human players in mind.
>
> 2) We are not sure on how to interpret this question. We assume it is referring to the actual time human players played to reach the world record, typically to the fact that some famous games as Pacman and Space Invaders received much more interest than the other Atari games by the gamers community. This indeed can lead to world records harder to beat than others, but we still think those records are much more representative of best human capabilities than the human baseline commonly used. If this interpretation is wrong, could the reviewer kindly provide more details so the authors can answer accordingly?
>
> 3) We choose to evaluate Rainbow as it is the current state-of-the-art on Atari and thus is one of the most important to re-evaluate. As mentioned we had limited amount of computational resources to conduct a full Atari benchmark with other DRL algorithms. As a side note, even Machado et al just re-evaluated on a single DRL algorithm, DQN. The second algorithm evaluated didn't imply DRL and therefore was much faster to train. Moreover, the focus of the article is first on measuring the impact of the diverging evaluation procedures, which is detailed in section 5.1 (applied to Rainbow), rather than re-ranking existing algorithms.

---

> > ### Comment · AnonReviewer3 · 2019-11-12
> > **Clarifying 2)**
> >
> > Thanks for your early response!
> >
> > To clarify 2), yes it refers to the time invested to reach the competency to set the world record.
> >
> > I very strongly agree that it is much more representative of human capabilities in terms of a human optimizing a comparable objective to the agent, and that the information is very valuable for the literature as many results are being misrepresented as "superhuman" game play. However, I am wondering if it could be a more suitable conclusion to use this information to instead argue that we shouldn't focus on comparing to humans at all. This is because:
> > - A considerable number of the scores maxed out, and it might not be very interesting to normalize scores by 999,999
> > - These scores were achieved over drastically different amounts of practice time, or like you acknowledged, attention from individuals willing to invest the time into setting records in these games. It's imaginable that for some of the games which have not maxed out, it is largely due to the communities not giving the game much interest, than being representative of what humans are ultimately capable of in the game.
> >
> > Perhaps this collection of results can instead be posed as an argument that we should only be comparing between algorithms, and perhaps choose a representative base algorithm (e.g., DQN) to report scores relative to?
> >
> > I hope that makes more sense, let me know if any of it is still unclear!

---

> > > ### Author Response · Authors · 2019-11-13
> > > **Why we think comparing to humans is still interesting**
> > >
> > > Thanks for this clarification, we understood better your point. However we still think comparing to humans can give useful information, particularly to know which range of score is effectively "good". We spend some times to look at the agent actually playing in order to understand what was going so wrong to create such difference between world record and state-of-the-art RL. We discovered that on many games (much more than the games usually coined $\textit{hard-exploration}$ games) the RL agent lacks of "common sens" and fails to understand anything of the game and this could be hidden if not relying on human score.
> > > Something which could be interesting would be to take the maximum possible score on each game as baseline. This could remove the problem of games receiving more attention by human players than others but we don't know if the maximum possible score is available on each game.
> > >
> > > Finally, the authors thinks that claiming superhuman performance is not only a scientific achievement, but has also the power to attract public interest beyond the scientific community.

---

### Official Review · AnonReviewer2 · 2019-10-23
**Official Blind Review #2**

**Rating:** 6

**Review:**

This paper revisits the way RL algorithms are typically evaluated on the ALE benchmark, advocating for several key changes that contribute to more robust and reliable comparisons between algorithms. It also brings the following additional contributions: (1) a new measure of comparison to human performance based on actual human world records (which shows that RL algorithms are not as « super-human » as is generally believed), and (2) an evaluation (based on the proposed guidelines) of Rainbow as well as a Rainbow-IQN variant (replacing the C51 component of Rainbow with Implicit Quantile Networks), showing that the latter brings a significant improvement upon the original Rainbow algorithm.

Overall I am leaning towards acceptance as I believe that such papers encouraging better benchmarking practice on Atari are definitely needed. Even if the technical contribution is limited, this paper could have a positive impact on the field by providing a clearer picture of the current state of deep RL algorithms on Atari (assuming that other researchers start following these recommendations -- and if that is not the case at least it will highlight issues with the way evaluation is currently done).

I do have a few concerns / questions though:

1.	I am not convinced by the recommendation to use performance during training for evaluation purpose. In Machado et al. (2018) it is argued that « this better aligns the performance metric with the goal of continual learning », but most deep RL algorithms trained on Atari games have not been intended to be used in a continual learning setting. It definitely has the advantage of being simple, but it seems to me that it can cause some issues, like making it difficult to compare different exploration techniques for off-policy learning (the exploration may cause poor behavior during training even if it helps the agent learn a better greedy policy), and more generally not being representative of the common practical use case where the goal is to obtain the best agent possible to use in production (with no further learning). Finally, it could make results even harder to reproduce due to the potential high variance of an agent’s performance at a fixed # of timesteps (vs. considering the max performance it can reach over the whole period). As a result, I am currently reluctant to see the proposed performance measure become the standard evaluation metric on ALE, and I would appreciate some additional justification from the authors on this point.

2.	Why not suggest to remove reward clipping in the recommendations? As mentioned in Section 6, reward clipping can prevent RL algorithms from properly playing some games, and thus in my opinion should be removed if the goal is to reach the highest score possible on all games. It seems to me that the choice of clipping the reward should be part of the algorithm (if it is not able to handle the high variety of « raw » rewards) and not of the benchmark environment, thus enabling further advancements towards algorithms that are robust to a wide range of rewards.

3.	Why bother to keep the mean performance when, as mentioned, it is highly sensitive to outliers compared to the median?


Additional remarks:
•	I might have missed it but I do not see the link to the source code. Am I correct to assume it will be released, to help with reproducibility?
•	It is not clear, when reading the paper, that the distributed version of Rainbow is actually constrained to mimic a single agent sequential algorithm in the experiments. I would suggest to remove mentions of the distributed version in the main text to avoid confusion, and mention it only in the Appendix section where it is used.
•	The « infinite reward loop » point at the end of Section 6 does not seem relevant in the list of reasons why Deep RL algorithms are far from the best human performance, since with infinite playtime and an infinite reward loop, the algorithm should be guaranteed to outperform humans.
•	I would have appreciated an evaluation of Rainbow-IQN with the current most commonly used evaluation schemes (e.g. the one used in the original Rainbow paper), for comparison purpose (even if such an evaluation has flaws, it is often the only performance measure available for existing deep RL algorithms)

Review update: thank you for the response, I am currently keeping my "Weak accept" rating because I agree it is important to highlight and (try to) fix the problems with the way algorithms are currently evaluated on ALE, in spite of the limited technical contributions (and the fact that I remained unconvinced regarding #1)

**Experience Assessment:**

I have read many papers in this area.

**Review Assessment: Checking Correctness Of Derivations And Theory:**

N/A

**Review Assessment: Checking Correctness Of Experiments:**

I carefully checked the experiments.

**Review Assessment: Thoroughness In Paper Reading:**

I read the paper thoroughly.

---

> ### Author Response · Authors · 2019-11-07
> **Answer to Reviewer #2**
>
> Thank you for your comments and feedback. We will try to answer to all your questions and remarks in the following.
>
> 1) Concerning the recommendation of reporting performance during training, we will base our argument on Machado et al. paper where they specifically speak about "Evaluation after learning" and "Evaluation of the best policy" (page 8). First we think that evaluating the best policy after learning hides both the data efficiency and the stability of the algorithm. Indeed most paper actually do not mention when the best results were encountered. Finally as stated in Machado et al. we think that "the best score achieved across training is a statistically biased estimate of an agent’s best performance". However, it could be interesting to report training performance along with a re-evaluation of the best model encountered while training.
>
> 2) We think that finding algorithms capable of managing high variety of rewards is still an open problem and most of algorithms are yet not suited to this. And as you mentioned clipping reward is an algorithmic choice, so we estimate it is out of scope of the recommendations of SABER. We think games on which reward clipping actually leads to sub-optimal policy are an important margin of improvement for algorithms trying to handle highly variable rewards.
>
> 3) We kept mean performance as most of previous works were reporting both median and mean. However, we think this is of limited interest and that is why all our graphics just plot the median normalized score.
>
> Additional remarks:
> •	The source code is currently available here: https://anonymous.4open.science/r/728e379d-4d38-49e2-9dd8-bf2fb4bd4844/
> •	We mentioned our distributed version of Rainbow as we think this give a good value on our source code even if like you mentioned we constrained it to mimic a single agent version in our experiments.
> •	We have reformulate this "Infinite reward loop" section in the revision we just submitted. The idea we wanted to highlight there was that agents are often stuck in a loop which in many cases is a sub-optimal behavior (in fact most of Atari games incorporate a timeout when not moving along the game and Elevator Action was the only one we found with an infinite reward loop).
> •	Unfortunately, we did not have time nor computational resources to run Rainbow-IQN following commonly used evaluation schemes.

---

### Public Comment · ~Rachit_Dubey1 · 2019-11-08
**Missing citations/relevant literature**

Hi authors,

I found your work to be very interesting and very timely. At the same time, I just want to point some relevant literature that you may want to add or cite in the paper. More specifically, some recent works (mentioned below) have compared human performance with RL algorithms and shown a gap between current deep RL approaches and human learning (specifically with regards to prior knowledge) and I believe that your paper would benefit by adding these papers. Thanks and good luck!

(a) Dubey, R., Agrawal, P., Pathak, D., Griffiths, T., & Efros, A. (2018). Investigating Human Priors for Playing Video Games. In International Conference on Machine Learning, ICML 2018 (pp. 1348-1356).

(b) Lake, B. M., Ullman, T. D., Tenenbaum, J. B., & Gershman, S. J. (2017). Building machines that learn and think like people. Behavioral and brain sciences, 40.

---

> ### Author Response · Authors · 2019-11-14
> **Citations inclusion**
>
> Thank you for your compliments and suggestions, these references are quite interesting. However, they focus more on the behavioral / cognitive approach of learning, which is not exactly in the scope of our article. Our article, although we mention the importance of fair comparison with human players, focuses on the evaluation of RL agents, but does not concern the learning process itself. For this reason, we think the references cannot be easily added to our related work or introduction.

---

### Decision · Program_Chairs · 2019-12-19

**Decision:**

Reject

**Comment:**

This paper proposes a new benchmark that compares performance of deep reinforcement learning algorithms on the Atari Learning Environment to the best human players.  The paper identifies limitations of past evaluations of deep RL agents on Atari. The human baseline scores commonly used in deep RL are not the highest known human scores.  To enable learning agents to reach these high scores, the paper recommends allowing the learning agents to play without a time limit.  The time limit in Atari is not always consistent across papers, and removing the time limit requires additional software fixes due to some bugs in the game software.  These ideas form the core of the paper's proposed new benchmark (SABER). The paper also proposes a new deep RL algorithm that combines earlier ideas.

The reviews and the discussion with the authors brought out several strengths and weaknesses of the proposal.  One strength was identifying the best known human performance in these Atari games.
However, the reviewers were not convinced that this new benchmark is useful.  The reviewers raised concerns about using clipped rewards, using games that received substantially different amounts of human effort, comparing learning algorithms to human baselines instead of other learning algorithms, and also the continued use of the Atari environment. Given all these many concerns about a new benchmark, the newly proposed algorithm was not viewed as a distraction.

This paper is not ready for publication. The new benchmark proposed for deep reinforcement learning on Atari was not convincing to the reviewers.  The paper requires further refinement of the benchmark or further justification for the new benchmark.